# Multi-Omic Analysis of Two Common P53 Mutations: Proteins Regulated by Mutated P53 as Potential Targets for Immunotherapy

**DOI:** 10.3390/cancers14163975

**Published:** 2022-08-17

**Authors:** Jayakumar Vadakekolathu, David J. Boocock, Kirti Pandey, Barbara-ann Guinn, Antoine Legrand, Amanda K. Miles, Clare Coveney, Rochelle Ayala, Anthony W. Purcell, Stephanie E. McArdle

**Affiliations:** 1John van Geest Cancer Research Centre, Nottingham Trent University, Nottingham NG11 8NS, UK or; 2Centre for Health, Ageing and Understanding Disease, School of Science and Technology, Nottingham Trent University, Nottingham NG1 4FQ, UK; 3Infection and Immunology Program, Biomedicine Discovery Institute, Melbourne, VIC 3800, Australia; 4Department of Biochemistry and Molecular Biology, Monash University, Melbourne, VIC 3800, Australia; 5Department of Biomedical Sciences, University of Hull, Hull HU6 7RX, UK

**Keywords:** p53, SaOS-2, R273H, R175H, mass spectrometry, MHC peptides, immunotherapy, conformational mutants

## Abstract

**Simple Summary:**

*TP53* is the most frequently mutated gene in many cancers, but it has failed to be a very effective target for treatment to date. To overcome this, we have examined what else changes in cells when the *TP53* gene is mutated. We modified cells that had no *TP53* expression to have one of the two most common mutations, either R175H or R273H. We examined how the presence of these *TP53* mutations caused cellular changes including microscopic, gene expression and peptide presentation to the immune system. This has allowed us to identify new (secondary) targets that could be used to facilitate the treatment of tumors that harbor p53 mutations.

**Abstract:**

The p53 protein is mutated in more than 50% of human cancers. Mutated p53 proteins not only lose their normal function but often acquire novel oncogenic functions, a phenomenon termed mutant p53 gain-of-function. Mutant p53 has been shown to affect the transcription of a range of genes, as well as protein–protein interactions with transcription factors and other effectors; however, no one has intensively investigated and identified these proteins, or their MHC presented epitopes, from the viewpoint of their ability to act as targets for immunotherapeutic interventions. We investigated the molecular changes that occurred after the *TP53* null osteosarcoma cells, SaOS-2, were transfected with one of two conformational p53-mutants, either R175H or R273H. We then examined the phenotypic and functional changes using macroscopic observations, proliferation, gene expression and proteomics alongside immunopeptidome profiling of peptide antigen presentation in the context of major histocompatibility complex (MHC) class I molecules. We identified several candidate proteins in both *TP53* mutant cell lines with differential expression when compared to the *TP53* null vector control, SaOS-V. Quantitative SWATH proteomics combined with immune-peptidome analysis of the class-I eluted peptides identified several epitopes presented on pMHC and in silico analysis shortlisted which antigens were expressed in a range of cancerous but not adjacent healthy tissues. Out of all the candidates, KLC1 and TOP2A showed high levels of expression in every tumor type examined. From these proteins, three A2 and four pan HLA-A epitopes were identified in both R175H and R273H from TOP2A. We have now provided a short list of future immunotherapy targets for the treatment of cancers harboring mutated *TP53*.

## 1. Introduction

The *TP53* gene is present in all cells and mutations within the gene represents the single most common alteration identified so far in most types of human cancers, which is, for some cancers, an early event. Point mutations, which represent the predominant mechanism by which p53 protein loses its normal tumor suppressor function, are located within the DNA binding domain of the protein, and 30% of them fall within what is called six “hotspot” residues, namely amino-acid residues R175H, G245S, R248Q, R249S, R273H and R282W, in various cancers. The frequency of TP53 mutations varies between the cancer types and is mostly mutated in cancer types such as uterine, lung, esophageal, head & neck, and pancreatic cancers [1]. Mutations, such as those affecting the six hotspot residues, either directly affect the p53 amino acid in contact with the DNA or affect the protein conformation of p53 hindering its wild-type function as a transcription factor and regulator of the cell cycle (recently reviewed in [2]).

DNA contact mutants have changes in the amino acids that make direct contact with DNA, leading to impaired p53-binding activity while sparing the intact p53 structure. On the other hand, conformational mutants robustly alter the p53 structure and disrupt the DNA binding activity [3]. Mutations of residues R175H, G245S, R249S and R282W affect the structural integrity of the p53 protein, linearizing the protein, whereas residues G245S, R248Q and R273H are directly involved with the DNA binding; these do not disturb the structure but will abolish the contact of the protein with its DNA target. It was concluded that the difference in structure between these two categories of mutants is at least partly responsible for the changes in phenotypes observed between the two groups of mutants. It is now understood that these mutations not only abolish p53 tumor suppressor function or antagonize the activity of the wild type p53 molecule, but they also give rise to mutated p53 proteins capable of actively promoting tumor development including invasive and metastatic properties [4,5,6]. Genetic variation arising from aneuploidy accounts for the diversity of *TP53* gain-of-function (GOF) phenotypes [7]. Not all *TP53* GOF mutants affect the same genes, and different p53 mutants use distinct mechanisms to promote cancer cell proliferation, invasion, metastasis and survival [8]. Indeed, Tan et al., 2015 [9] demonstrated that the mutation occurring at residue p53-R273H (a contact mutant), but not the conformational mutant p53-R175H, promotes cancer cell survival and resistance to cell death (anoikis) of the cancer cell via constitutively activating PI3K/AKT signaling. Moreover, different mutants appear to affect p53 localization, which, in turn, has been shown to have an opposite effect on autophagy. It appears that mutated p53 proteins located in the nucleus promote autophagy, while those located in the cytoplasm inhibit it [10].

In addition, the p53 mutation at amino-acid residue R273H has been linked with inflammation; a pancreatic cell line harboring this mutation was shown to have prolonged NF-κB activation through TNF-α signaling [11]. Many of the genes affected by these two classes of mutants have been identified (reviewed in [12]), but their potential usefulness as targets for cancer immunotherapy has received limited attention [13]. Since the two *TP53* mutants (R175H and R273H) directly influence protein–protein interactions in different ways, we investigated the molecular changes that occur as a direct consequence of the presence of these two p53 mutants in otherwise p53 null cells. To do this, we used the osteosarcoma cell line, SaOS-2, which lacks any endogenous *TP53* expression. Correlations between gene expression changes, protein levels and peptide presentation on MHC class I were determined in a multi-omics approach.

## 2. Materials and Methods

### 2.1. Cell Lines

SaOS-2 osteosarcoma cell lines [14] transfected with pBR322 vector control, *TP53* R175H or R273H mutants [15] were grown in DMEM medium (Gibco, Waltham, MA, USA) containing 10% Foetal Calf Serum (Gibco), 1% L-Glutamine (Lonza, Basel, Switzerland) and 1 mg/mL G418 (Sigma Aldrich). All the cells were grown in a 37 °C humidified incubator with 5% CO_2_ and sub-cultured at 80–90% confluence. *TP53* mutation status was confirmed in the transfected cell lines using PCR and Sanger sequencing of products amplified using primers flanking the *TP53* mutations (forward primer 5′ TCTTCTGTCCCTTCCCAGAA3′ and reverse primer 5′ CAAGGCCTCATTCAGCTCTC3′) and cDNA template synthesized (Promega, Madison, WI, USA) using RNA isolated (Qiagen, Hilden, Germany) from the transfected cell lines.

### 2.2. Cell Adhesion Assays

Cell adhesion assays were performed using Collagen I-coated plates (Gibco). Briefly, cells were trypsinized and counted using a hemocytometer, cells were washed twice using serum free media and resuspended in 10 mL of serum free media at a count of 100,000 cells and were seeded into each well of the coated plates. The plate was then incubated for 3 h at 37 °C in a 5% humidified CO_2_ incubator. After 3 h, the media were removed from each well, and the cells were washed twice with PBS. After the final wash, the cells were fixed with 4% paraformaldehyde at room temperature. The wells were washed twice with PBS and were stained by adding 50 µL of crystal violet/cell stain solution and incubated at room temperature. The wells were washed twice with deionized water and dried at room temperature. The plates were scanned using a C.T.L. ELISPOT plate reader and the number of remaining cells counted using ImmunoSpot software (CTL Ltd., Shaker Heights, OH, USA).

### 2.3. Colony Forming Assays

Single cell suspensions were plated at the clonal density of 125 cells per well in a 6-well plate, 2 wells per clone. Following 10 days in culture, the colonies were fixed with 4% (*w*/*v*) paraformaldehyde for 15 min at 4 °C and stained with crystal violet solution (0.5% (*w*/*v*) crystal violet in 70% (*v*/*v*) ethanol) for 15 min at room temperature, after which the colonies were washed with DPBS and allowed to dry prior to counting under a light microscope. Colonies with fewer than 32 cells were excluded from the counts. The plates were then photographed. These experiments were conducted in duplicate wells in three independent experiments.

### 2.4. Proliferation Assays

Proliferation rates of the two different *TP53* mutant cell lines were investigated in comparison to the vector control cell line using 3-(4,5-dimethylthiazol-2-yl)-2,5-diphenyltetrazolium bromide (MTT) dye reduction assay (Sigma). Briefly, 3000 cells were seeded in the wells of 96-well plates in triplicates in complete DMEM medium, and the cells were allowed to grow for 48 h. MTT reagents were added to each well and incubated for 2 h in the cell culture incubator. After 2 h, the culture media was carefully removed and 150 μL of DMSO added. The plates were incubated on a shaker for 10 min, and the absorbance was measured at 570 nm.

### 2.5. Gene Expression Arrays

RNA was isolated and labelled as described previously [16] and hybridized on to Agilent Human 44K arrays. All the analyses including the QC were performed using Agilent Genespring V.11. Fold change calculations between the groups were performed using a volcano plot with multiple testing correction, and a Benjamini–Hochberg-corrected *p* value of 0.01 and a fold change of ≥3.0 were considered significant. *TP53* expression at the transcript level was confirmed using SyBR green qRT-PCR using TP53 forward (TGAAGCTCCCAGAATGCCAG) and TP53 reverse (GCTGCCCTGGTAGGTTTTCT) primers, Human glucuronidase beta (GUSB, Forward primer: ACTGAACAGTCACCGAC, Reverse primer: AAACATTGTGACTTGGCTAC) was used as housekeeping normalizer, and mRNA expression was calculated using ΔΔCT method.

### 2.6. Cell Lysis and Homogenization

Cells were cultured up to 80% confluence in a T175 flask. Cells were washed thrice with ice-cold PBS, and the residual liquid was removed completely. Cells were lysed subsequently by adding 500 µL of lysis buffer (9.5 M urea + 2% Dithiothreitol and 1% N-Octyl-Beta-Glucopyranoside) with MS-SAFE Protease and Phosphatase Inhibitor (Sigma). The crude lysate was collected into microcentrifuge tubes and further lysed by a handheld micro-pestle. All the samples were subjected to three cycles of sonication with 5 min intervals on ice. Debris was removed by centrifugation at 10,000× *g* for 15 min at 4 °C. One in ten dilution of the final solution was used to determine the protein quantity using a protein assay dye kit (BioRad, Hercules, CA, USA). Samples of 50 µg of the total proteins were trypsinized overnight at 37 °C using trypsin in accordance with the manufacturer’s instruction. After trypsinizations, samples were dried using a vacuum concentrator and resuspended in 5% acetonitrile in 0.1% formic acid for mass spec analysis.

### 2.7. Liquid Chromatography with Tandem Mass Spectrometry (LC MS/MS) Analysis (IDA/SWATH)

Samples (~50 µg protein) were reduced, alkylated and prepared as previously described [17]. Samples (3 µL, ~5 µg protein digest) were directly injected in triplicate by autosampler (Eksigent nanoLC 425 LC system) at 5 µL/min onto a YMC Triart-C18 column (15 cm, 3 µm, 300 µm i.d.) using gradient elution (2–40% mobile phase B, followed by wash at 80% B and re-equilibration) over 60 min (80 min run time) for spectral library construction using data-/information-dependent acquisition DDA/IDA and SWATH. Mobile phases consisted of A: water containing 0.1% (*v*/*v*) formic acid; B: acetonitrile containing 0.1% (*v*/*v*) formic acid. The LC system was hyphenated to a Sciex TripleTOF 6600 mass spectrometer fitted with a Duospray source and 50 µm electrode suitable for microflow proteomic analysis. The IDA method was run with parameters of: CUR 25; GS1 12; GS2 0; ISVF 5500; TEM 0. TOFMS mass range of 400–1250 *m*/*z*; accumulation time of 250 ms with product ion scans on the top 30 ions before switching (dynamic exclusion for 20 s) with rolling collision energy selected. Product ion accumulation time was set to 50 ms, giving a cycle time of 1.8 s. The SWATH method source parameters were CUR 25; GS1 15; GS2 15; ISVF 5500; TEM 0 with a 65 ms TOFMS scan followed by 40 variable SWATH windows (optimized on an IDA datafile of a relevant sample) of 65 ms between 399.5 and 1593 *m*/*z*, giving a cycle time of 2.7 s.

### 2.8. SWATH Data Analysis

In total, 45 SWATH data files (3 groups, 5 biological replicates, 3 technical replicates) were processed for quantitation using DIANN 1.8 utilizing a library free search (Swissprot human Jan 2022 fasta), a deep learning approach, as described in Demichev et al. [18]. Differential protein expression was carried out using the Limma R package as exposed by the R package “StatsPro” [19].

### 2.9. Purification of Antibodies and Preparation of Immunoaffinity Columns

Human leukocyte antigen (HLA) antibodies specific to HLA-A2 (BB7.2 [HB-82; ATCC] [20] and pan-HLA class I (W6/32 [HB-95; ATCC] [21] were purified from hybridoma cells. Secreted monoclonal antibodies, BB7.2 (anti-HLA-A2) and W6/32 (pan-HLA class I), were harvested from spent media and purified using Protein A Sepharose (PAS, CaptivA^®^, Repligen, Waltham, MA, USA) using a Profinia purification system (Biorad, Hercules, CA, USA). Next, an affinity matrix wherein 10 mg/mL of BB7.2 (anti-HLA-A2) and W6/32 (pan-HLA class I) were crosslinked to 1 mL of PAS resin were prepared for each of the transfected SaOS cell line, as previously described [22].

### 2.10. HLA Purification and Peptide Elution

Snap-frozen pellets of 5 × 10^8^ cell were obtained for each transfected SaOS cell line. Immunoaffinity capture of solubilized peptide HLA (pHLA) was performed as described previously [23]. Briefly, cell pellets were pulverized by cryogenic milling (Retsch Mixer Mill MM 400) followed by detergent-based lysis using 0.5% IGEPAL (Sigma-Aldrich, St. Louis, MO, USA). The lysates were cleared by ultracentrifugation, and the supernatant was passed through a PAS pre-column to remove any non-specific binders, followed by serial affinity pull down of HLA-A*02:01 peptide complexes using BB7.2 and the remaining HLA class I allotype complexes using W6/32 columns. pHLA complexes were eluted using 10% acetic acid and fractionated by reversed-phase high-performance liquid chromatography (RP-HPLC) on a 4.6 mm internal diameter × 100 mm monolithic reversed-phase C18 HPLC column (Chromolith SpeedROD; Merck Millipore, Darmstadt, Germany) using an ÄKTA micro HPLC (GE Healthcare, Chicago, IL, USA) system using a mobile phase of Buffer A (0.1% Trifluoroacetic acid) and Buffer B (80% Acetonitrile, 0.1% trifluoroacetic acid). Fractions containing peptide were placed into 20 pools, vacuum concentrated (LABCONCO, Kansas City, MO, USA) and reconstituted in 0.1% formic acid.

### 2.11. Analysis of HLA Class I Bound Peptides by LC-MS/MS

Reconstituted fraction pools were spiked with 11 iRT peptides [24] and were analyzed by LC-MS/MS via a data dependent acquisition strategy using a NanoUltra cHiPLC system (Eksigent) coupled to an SCIEX 5600+ TripleTOF mass spectrometer equipped with a Nanospray III ion source. Samples were loaded onto a pre-equilibrated cHiPLC trap column (3 µm, ChromXP C18CL, 120 Å, 0.5 mm × 200 µm) at 5 µL/min in buffer A (0.1% formic acid, 2% acetonitrile) over 10 min and separated over a cHiPLC column (3 µm, ChromXP C18CL, 120 Å, 15 cm × 75 µm) using a linear gradient starting from 2% buffer B (80% acetonitrile, 0.1% formic acid) over 100 min at a flow rate of 300 nL/min for 1 min, 2–35% buffer B for 75 min, 35–80% for 4 min followed by 10 min of 80% buffer B, decreasing to 2% B for 10 min. of 2–80% Buffer B /Buffer A (0.1% formic acid). Data acquisition occurred with the following instrument parameters: ion spray voltage, 2400 V; curtain gas, 30 l/min; ion source gas, 20 l/min; and interface heater temperature, 150 °C. MS/MS switch criteria selected the top 20 ions meeting the following criteria per cycle: *m*/*z* >200 amu, charge state of +2 to +5, intensity >40 counts per second (cps). After two selections for fragmentation, ions were ignored for 30 s.

### 2.12. Immunopeptidomic Data Analysis

MS/MS spectra in Wiff files were searched against the human proteome (Uniprot 15/06/2017; 20,182 entries) using Peaks X Pro software. The following search parameters were used: error tolerance of 15 ppm using monoisotopic mass for precursor ions and 0.1 Da tolerance for fragment ions; enzyme use was set to none with following variable modifications: oxidation at Met, deamidation at Asp and Gln. False discovery rate (FDR) was estimated using a decoy fusion method [25] and all datasets were analyzed at 5% FDR.

For data filtering, first, iRT peptides were removed along with any duplicated peptide sequences. Secondly, only peptides of 8–15 amino acid in length were retained. To define the length distribution and global peptide binding motif including presence of specific residues (primary anchor positions at P2 and PΩ (the C-terminal reside)) for different alleles and mutants, non-redundant peptides were considered for the immunopeptidome dataset. Length distribution data was plotted and visualized using GraphPad Prism version 9.1.0 (GraphPad Software, USA, www.graphpad.com (accessed 1 May 2022)), and motifs were visualized using Icelogo software [26] with the percentage difference when compared to the human proteome being performed using a static reference method. Binding affinity was determined using NetMHC Pan 4.0 [27]. The mass spectrometry immunopeptidomics data were deposited in the ProteomeXchange Consortium [28] using PRIDE [29]. The project accession number for the dataset is PXD034044.

### 2.13. In Silico Analysis of the Identified Proteins Indicated Their Expression in a Broad Spectrum of Healthy Tissues and Cancer

Human Protein Atlas version 19.3 was used to further narrow down the list of potential targets based on their expression in tumor tissues compared to normal tissue counterparts.

## 3. Results

### 3.1. The Two Mutants of TP53 Had Significantly Different Phenotypical, Functional and Gene Expression

In order to study the downstream genes/proteins regulated by p53 mutants, SaOS-2 cells, which had both *TP53* alleles deleted [30], were transfected with either the pcDNA plasmid alone to provide an empty plasmid vector control, which we named SaOS-V. Alternatively, the SaOS-2 cells were transfected with pcDNA plasmid encoding the conformation p53 mutant R175H to create SaOS-R175H or a DNA contact p53 mutant R273H to create SaOS-R273H, as described previously [17]. A change in morphology was observed (Figure 1A) between the two mutants and the control, with SaOS-R273H showing a more fibroblast-like phenotype. The success of the transfections was confirmed by PCR (Figure 1B) and the single-point mutation by sanger sequencing (Figure 1C). It was also observed that SaOS-R273H formed more colonies in colony-forming assays (Figure 1D,F), demonstrated superior cell adhesion properties (Figure 1E) and showed a significantly faster proliferation rate (Figure 1G) than SaOS-R175H.

mRNA array profiling showed that 199 genes were significantly upregulated (linear FC ≥ 5 and using an adjusted *p* value for multiple comparison of 0.01) and 360 genes downregulated in SaOS-R175H when compared with the vector control. Similarly, 201 genes were significantly upregulated, and 131 genes downregulated in SaOS-R273H, compared to the SaOS-V vector control. In total, 34 genes were commonly upregulated, and 41 genes were commonly downregulated in the SaOS-*TP53* mutants compared to the SaOS-V vector control (Figure 1H,I; Appendix A). The top 40 up- and downregulated genes in each mutant were identified (Figure 1J,K) following a comparison with gene expression in SaOS-V cells. Genes that were uniquely upregulated were then further analyzed using the Metascape pathway enrichment tool (Figure 1L,M). The most upregulated pathways in SaOS-R175H mutants compared to vector control were integrin binding (Log P = −5.44), receptor regulator activity (Log P = −4.28) and epithelial-to-mesenchymal transition (EMT) (Log P = −3.43). While hotspot *TP53* mutants have been reported to contribute to the chronic inflammation that underlies the development of cancer through cooperation with NF-κB, we have found that SaOS-R273H regulated more inflammation-related genes than SaOS-R175H, including hallmarks of TNF-α (Log P = −4.18) and IL-2 STAT-5 signaling (Log P = −1.9) (Figure 1L,M). A complete list of the pathways enriched in up- and down-regulated genes in both SaOS-*TP53* mutants can be found in Appendix A.

### 3.2. Quantitative SWATH Proteomics

Quantitative SWATH proteomics showed that SaOS-R175H and SaOS-R273H mutants expressed different proteins in comparison to each other and the *TP53* null vector control. Figure 2A shows the steps followed. In total, 481 proteins were uniquely upregulated in SaOS-R273H mutants compared with the null vector, while 318 were identified as uniquely upregulated in the SaOS-R175H mutant (Figure 2B). Confirmation and correlation of the protein level of p53 expression assessed by SWATH-MS and gene expression by qRT PCR in the mutants versus null vector are shown in Figure 2C. SaOS-R273H expresses P53 at a higher level than SaOS-R175H (Figure 2C), which correlates with the mRNA data (Figure 1B).

Label-free SWATH proteomics identified 4476 proteins in total in the three cell lines. Differential expression analysis revealed that 2278 proteins were differentially expressed in the SaOS-R273H mutant compared to vector control (≥0.3 log2 fold change, P. adjust = 0.05), of which 1110 proteins were upregulated (Figure 2D, only proteins ≥ log2 fold change above 1 shown), whereas 1913 proteins were differentially expressed between SaOS-R175H cells when compared to the SaOS-V vector control (≥0.3 log2 fold change, P. adjust = 0.05), of which 947 proteins were upregulated (Figure 2E, only proteins ≥ log2 fold change above 1 shown). A complete table of differentially expressed proteins is available in Appendix A. Pathway enrichment analysis (Metacore™) of the upregulated proteins in both SaOS-*TP53* mutant cell lines revealed that completely different pathways were affected in the two SaOS-*TP53* mutants (Figure 2F,G; Appendix A). The pathways that were upregulated in SaOS-R175H cells included those involved in immune response antigen presentation (FDR = 1.12^−5^) and ubiquitin-proteasomal proteolysis (FDR = 1.12^−5^) (Appendix A).

Similarly, the major pathways that were upregulated in SaOS-R273H compared to SaOS-V were transcription mRNA processing (FDR = 6.79^−17^), protein folding (FDR = 7.92^−15^), translation initiation (FDR = 7.2^−10^) and response to unfolded proteins (FDR = 5.42^−9^) (Appendix A).

### 3.3. HLA Immunopeptidome Analysis Identified Unique and Shared Peptides between the SaOS-TP53 Mutants

The workflow for the identification of MHC-bound peptides is shown in Figure 3A. pHLA complexes for SaOS-R175H and SaOS-R273H were isolated by immunoaffinity purification using the HLA specific antibodies; BB7.2 for capture of HLA A*02:01-peptide complexes followed by W6/32 for capturing all other HLA class I allotype-peptide complexes. Analysis of the MHC peptide repertoire of the SaOS-R175H and SaOS-R273H cell lines revealed several unique HLA-A2 (Figure 3B) and pan-HLA-derived peptides (Figure 3C) (for a complete list of HLA-A2 derived peptides see Appendix A; for a complete list of MHC class-I derived peptides, see Appendix A).

In the SaOS-R175H cell line, a total of 18,941 peptides (Table 1) were identified with 9127 and 9814 peptides identified using BB7.2 and W6/32 antibody, respectively. There were 2112 peptides overlapping between the two datasets (Appendix A). Similarly, in the SaOS-R273H, a total of 10,115 peptides (Table 1) were identified with 2970 and 7145 unique peptides identified in HLA-A2 and pan HLA, respectively, and only 746 peptides overlapping between the two (Appendix A).

As anticipated, peptides identified in both *TP53* transfectants including HLA-A*02:01 and pan class I exhibited typical class I length distribution, with most peptides (67–70%) being nonamers followed by 10 mers and 11 mers (Appendix A). The nonamers were used from each dataset to construct peptide binding motifs. The HLA-A*02:01 peptides identified in SaOS-R175H and SaOS-R273H matched the expected HLA-A*02:01 binding motif with small hydrophobic amino acids at P2 and P9 (Appendix A). Allele-specific binding motifs from the W6/32 immunoprecipitated material were determined by ranking allele-specific binding affinity for all nonamers using NetMHC Pan 4.0 [27] for the remaining HLA allotypes expressed by the SaOS variants. Peptides with a rank threshold of less than 0.5 were considered as strong binders, whilst those between 0.5–2.0 were considered as weak binders. The HLA-binding predisposition data was then used to plot allotype-specific binding motifs for the remaining HLA class I alleles using Icelogo (Appendix A).

Furthermore, we investigated p53 peptides identified from the two SaOS-*TP53* mutant cell lines. Nine and seven unique p53-specific peptides were found in SaOS-R175H and SaOS-R273H, respectively (Appendix A). Of note, unique peptides proximal to the reciprocal mutation were found in the two transfectants, i.e., P53_264–272_ (LLGRNSFEV) was found only in SaOS-R175H transfectant, whilst peptide P53_164–172_ (KQSQHMTEV) was found only in SaOS-R273H transfectant. This suggests that the mutations impact antigen processing by the SaOS-*TP53* variants.

Number of proteins contributing to the immunopeptidome varied between the two transfectant cell lines (Table 1; Appendix A). With the quantitative global proteomics data, we have identified 947 upregulated proteins in the SaOS-R175H and 1110 proteins in SaOS-R273H cell lines. These genes/proteins were compared against the identified HLA-A2 (BB72) and Pan HLA (W6/32) peptidome of the corresponding cell lines yielding 190 potential HLA-A2 protein targets in SaOS-R175H and 101 potential HLA-A2 protein targets in SaOS-R273H (Figure 3B,C), Similarly, 201 potential Pan-HLA protein targets in SaOS-R175H and 193 potential Pan-HLA protein targets in SaOS-R273H. We then looked at the common protein targets between the mutants in both HLA-A2 fraction and Pan HLA fractions. In total, 50 proteins were found to be common between the mutants in the HLA-A2 fraction, and 87 proteins were found to be common between the Pan HLA fractions. These common proteins were used to identify common candidates shared between both the HLA fractions (Pan and A2) and identified 34 common proteins (Figure 3F). We have further investigated these proteins at peptide level; Figure 3G shows the number of epitopes identified from each short-listed protein and in both mutants using both BB72 and W6/32 antibodies. Peptide level analysis (Figure 3H) revealed 18 unique epitopes in pan273, 29 unique epitopes in pan175, 9 unique epitopes in HLA-A2 273 and 42 unique epitopes in HLA-A2 175 mutants. A total of 19 epitopes were common to all four fractions.

### 3.4. In Silico Analysis of the Identified Peptides Revealed Targetable Peptides in TP53 Mutants

To further narrow down the list of potential targets, the expression of these proteins was analyzed using the Human Protein Atlas version 19.3 (Figure 4). The utility of the final candidates was confirmed again by assessing their expression in tumor tissues compared to normal tissue counterparts (Figure 5). From the shortlisted 34 protein candidates, we have further narrowed the top 10 candidates that had minimum expression on the normal tissues for the final list. These proteins are KCL1, UHCL1, MAP1A, LAMB2, FLNC, NOP2, HEXB, PALLD, GLRX3 and TOP2A. Further analysis of the above proteins using the Human Protein Atlas in 20 cancer types showed that most of them have overexpression in cancer tissues compared to the normal counterparts (Figure 5). KLC1 and TOP2A showed high levels of expression in almost every tumor type examined (Figure 5), while UCHL1 was found mainly in glioma. The remaining protein candidates showed varying degrees of expression in cancer tissues (Figure 5).

## 4. Discussion

The longer half-life of mutant p53 compared to the wild-type protein and its frequent mutation in cancer renders mutant p53 a good target for immunotherapy. The majority of *TP53* mutations are either DNA-contact or conformation mutant type. DNA-contact mutants lose their function because the mutation abolishes the binding of p53 to its DNA target. In contrast, conformational mutants cause an opening/linearization of their sequence due to the loss of a crucial amino acid needed to maintain the 3D fold (reviewed by [31]), which is also thought to prevent immune recognition. Researchers have since shown that this loss of tumor suppression was accompanied by a gain in oncogenic functions such as invasion, metastasis and chemoresistance [32]. This is achieved by *TP53* mutants because, apart from the point mutation, the function of their other domains remains unaltered. Interestingly, while Prives’s group showed that the R175H, R248Q and R273H mutants affected the most frequently altered amino acid residues, they have also shown that not all mutations of *TP53* were created equal, with some mutants exerting GOF effects and others abolishing wild-type p53 activity [3].

P53-derived vaccines have not produced the anticipated clinical outcome. Indeed, no significant reduction in tumor burden has been observed in most clinical trials to date, despite the detection of p53-specific vaccine-induced immunological responses [33,34]. This can be explained by the difficulties in targeting a loss of function and trying to target every possible mutation when there is such a diversity [35]. However, eight mutations account for one-third of all *TP53* mutations found in tumors, suggesting some homogeneity due to these mutations preferentially occurring because of environmental factors, or being the most impactful in a range of tissue types, occurring by virtue of the features of the sequences in these regions, i.e., CpG sites or because they are GOF-tumor-promoting mutations (recently reviewed in [36]). It is most likely to be a combination of all of these factors. Currently, the most promising *TP53*-targetting therapies involve blocking the degradation of the wild type protein using small molecules and stapled peptides [37], the reactivation of mutant forms of *TP53* with wild type properties [36] and, increasingly, the development of mutant p53-specific bispecific antibodies, which are now entering pre-clinical trials [38].

To circumvent the issues that could arise due to the off-targeting of healthy cells that express the ‘Guardian of the Genome’ [34], we wanted to identify secondary targets, genes and proteins whose expression/levels were elevated by two of the most commonly occurring p53 mutations, R175H and R273H, each of which impact the wild-type function of p53 in different ways. R175H is the most frequent of all p53 mutations, accounting for 5.6% of cases, while R273H is the third most frequent, accounting for 3.95% of cases [12]. Each was chosen to represent two key impacts of mutations on p53 function, p53 conformation and DNA contact, respectively. We transfected SaOS-2 cells, which innately lack *TP53* expression, with plasmids carrying either *TP53* mutant. Any changes in global gene and protein expression were measured in comparison to vector control transfected SaOS-2 cells. R175H mutations turned on mostly integrin binding genes, receptor-regulator activity and epithelial mesenchymal transition genes, while R273H mostly turned on TNFα signaling via NF-kB, IL-2 and STAT5 signaling and hallmarks of inflammatory responses. These had been reported effects of the R175H [6,39,40] and R273H mutations [11,41,42] previously.

We evaluated the immunopeptidome of the two *TP53*-transfected cell lines. We have identified a total of 18,941 peptides in the SaOS-R175H cell line and a total of 10,115 peptides in the SaOS-R273H cell line.

Consistent with changes in transcriptomics and proteomics, a number of unique peptides were found presenting in the two different SaOS-*TP53* mutant cell lines. Of note, the position of the two point mutations in *TP53* also altered antigen processing of the p53 protein itself, resulting in nine and seven unique peptides identified in the SaOS-R175H and SaOS-R273H transfectants, respectively. Intriguingly, unique peptides proximal to the reciprocal mutation were found in the two transfectants, i.e., P53_264–272_ (LLGRNSFEV) was found only in SaOS-R175H transfectant, whilst peptide P53_164–172_ (KQSQHMTEV) was found only in SaOS-R273H transfectants, suggesting the mutations impact on antigen processing and/or presentation of the two p53 proteins. Thus, *TP53* mutation not only generates unique peptides that may form targets for vaccination or immunotherapy but also forms distinctive signature peptides that distinguish between the two mutant *TP53* variants.

We found overlap between the large number of proteins expressed by each SaOS variants and MHC eluted peptides confirmed the processing and presentation of common epitopes on MHC class-I. These proteins were then examined for their expression in 97 healthy tissues and 20 different tumor types. TOP2A and KLC1 were the most frequently expressed of the upregulated protein in SaOS mutants, overexpressed in many cancers with TOP2A and expressed in many healthy tissues.

Kinesin family members are regulators of mitosis and cytokinesis [43] playing key roles in tumor development and progression. Already successfully targeted by microtubule inhibitors and drugs [44], the issue remains that the microtubule-based cytoskeleton is essential for a number of fundamental biological processes including mitosis [43]. This situation has spurred efforts to screen and target cancer-specific kinesins, for which inhibition hampers mitosis without producing significant side effects. It is therefore essential to identify cancer specific kinesins. KLC1 is a microtubule-associated force-producing protein that is thought to play a role in organelle transport. KLC1’s role in breast cancer, as a suppressor of epithelial-mesenchymal plasticity and translocation to the 5′ end of anaplastic lymphoma kinase (ALK), has been described previously [45]. Indeed, ALK inhibitors, such as crizotinib, have been shown to have therapeutic effects in clinical trials for lung cancer (systematically reviewed in [46]), inflammatory myofibroblastic tumors, a rare mesenchymal neoplasm [47] and anaplastic large-cell lymphoma [48]. Of note, KLC1 is a favorable prognostic marker in pancreatic cancer (*p* < 0.001) and an unfavorable prognostic marker in renal cancer with highest median expression in gliomas (TCGA dataset). KLC1 was undetectable in most healthy tissues but found in almost all tumor tissues with highest levels in the brain and moderate to high levels in the cytoplasm of almost all tumors tested by immunohistochemistry (Human Protein Atlas; http://www.proteinatlas.org; [49]).

## 5. Conclusions

Further preclinical studies will involve the validation of the targets identified in this study. We will confirm the immunogenicity of the peptides we have shown to be presented in the context of MHC, in vivo, and validate their potential to induce anti-tumor effects in clinical trials. However, in this study, we have identified a number of targets and their presented epitopes, and we have validated their cancer specificity in comparison to a large cohort of healthy tissues, providing identification of new secondary targets for the treatment of p53-positive tumors.

## Figures and Tables

**Figure 1 cancers-14-03975-f001:**
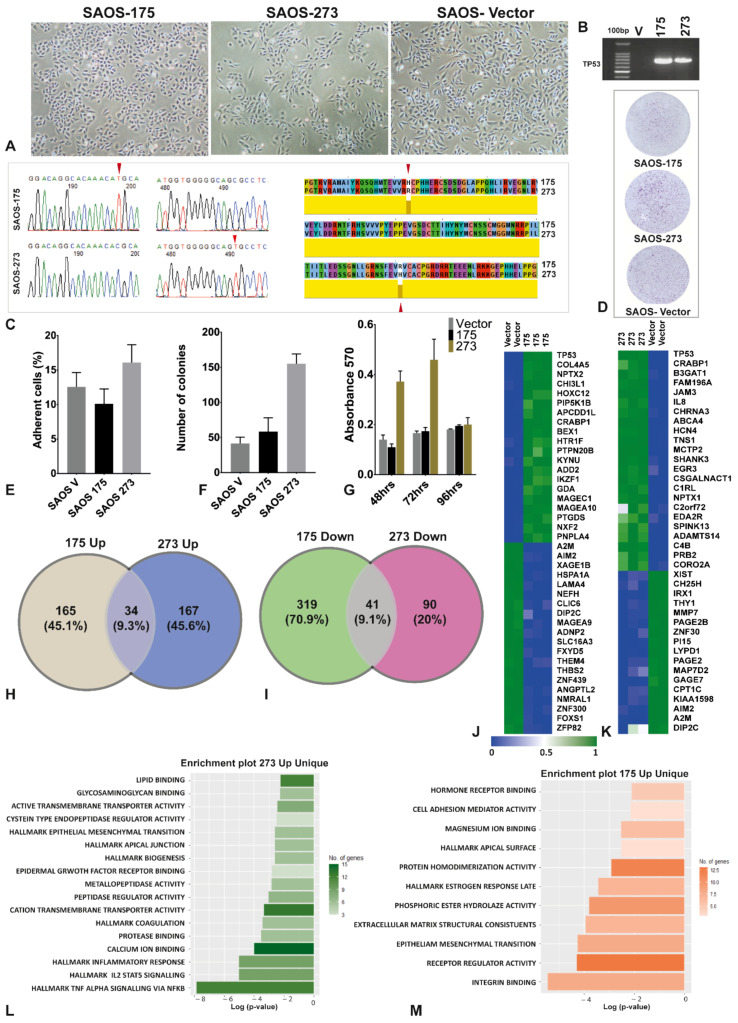
SaOS-p53 mutants showed significant differences at the phenotypical, functional, and molecular levels. (**A**) Light micrograph (5× magnification) showing the morphology of the SaOS-V, SaOS-R175H and SaOS-R273H variants; (**B**) agarose gel electrophoresis of amplicons of the two mutated *TP53* and the null vector confirmed expression of the transfected material in SaOS-R175H and SaOS-R273H cells; (**C**) BlastX alignment of Sanger sequencing results showing the p53 mutations at the amino acid level; (**D**) representative images of the three *TP53* mutant cell lines following a colony forming assay; (**E**) bar graph showing the results of cell adhesion assay (mean and SEM, n = 3), Y-axis represent % of adherent cells in each mutant and control; (**F**) bar graph showing the results of the colony-forming assay (mean and SEM, n = 3); (**G**) bar graph of proliferation assay conducted using MTT assay at three time points (mean and SEM, n = 3); (**H**,**I**) Venn diagram showing the common genes up and downregulated in R175H and R273H mutants compared to the null vector; (**J**,**K**) heatmaps of the 50 most significantly up and downregulated genes in both R175H and R273H compared to null vector control; (**L**,**M**) metascape gene enrichment analysis of the upregulated pathways in R175H and R273H mutant cell lines, X-axis represents log (*p*-value), and y-axis represents significantly enriched pathways.

**Figure 2 cancers-14-03975-f002:**
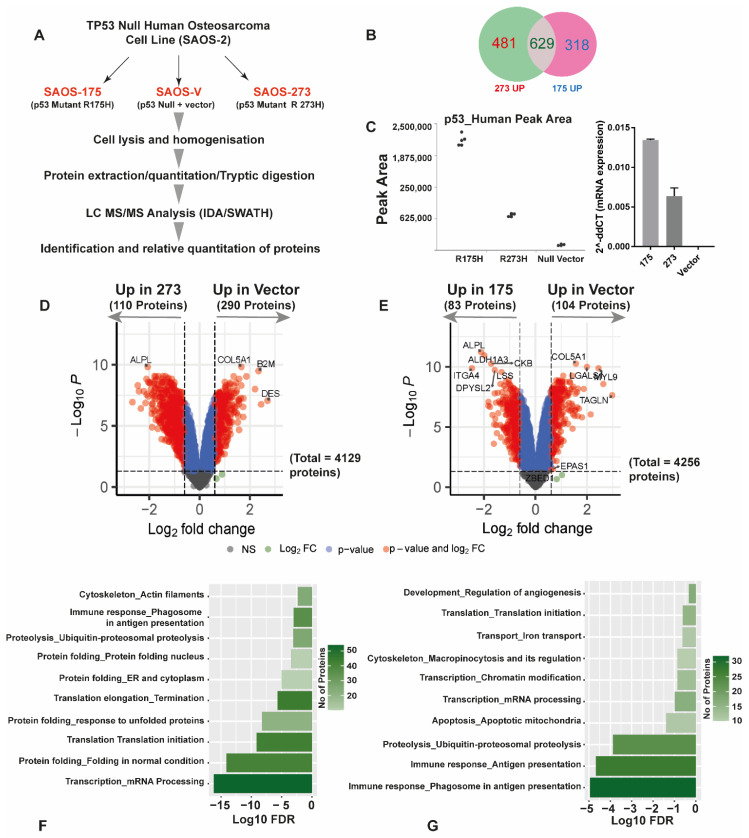
Label-free global quantitative SWATH proteomics revealed significant proteome changes in the SaOS-R175H and SaOS-R273H cell lines compared to SaOS-V. (**A**) Workflow followed for generating quantitative proteomics data; (**B**) Venn diagram showing common and unique proteins differentially upregulated in R273H and R175H mutants compared to vector control; (**C**) P53 levels in each cell line by mass spectrometry and qRT PCR; (**D**) volcano plot showing differentially expressed proteins in SaOS-R273H compared to SaOS-V cells (significant proteins are highlighted in solid red log2 fold cutoff ≥ 1). The X-axis indicates log2 fold change and Y-axis indicates ^−^log10 *p* value; (**E**) volcano plot showing differentially expressed proteins in SaOS-R175H compared to SaOS-V. Significant proteins were highlighted in solid red (log2 fold cutoff ≥ 1); (**F**,**G**) metacore enrichment analysis of the upregulated pathways in the SaOS-R175H and -R273H variants. X-axes represent log 10 FDR, and y-axes represent the different significantly enriched pathways.

**Figure 3 cancers-14-03975-f003:**
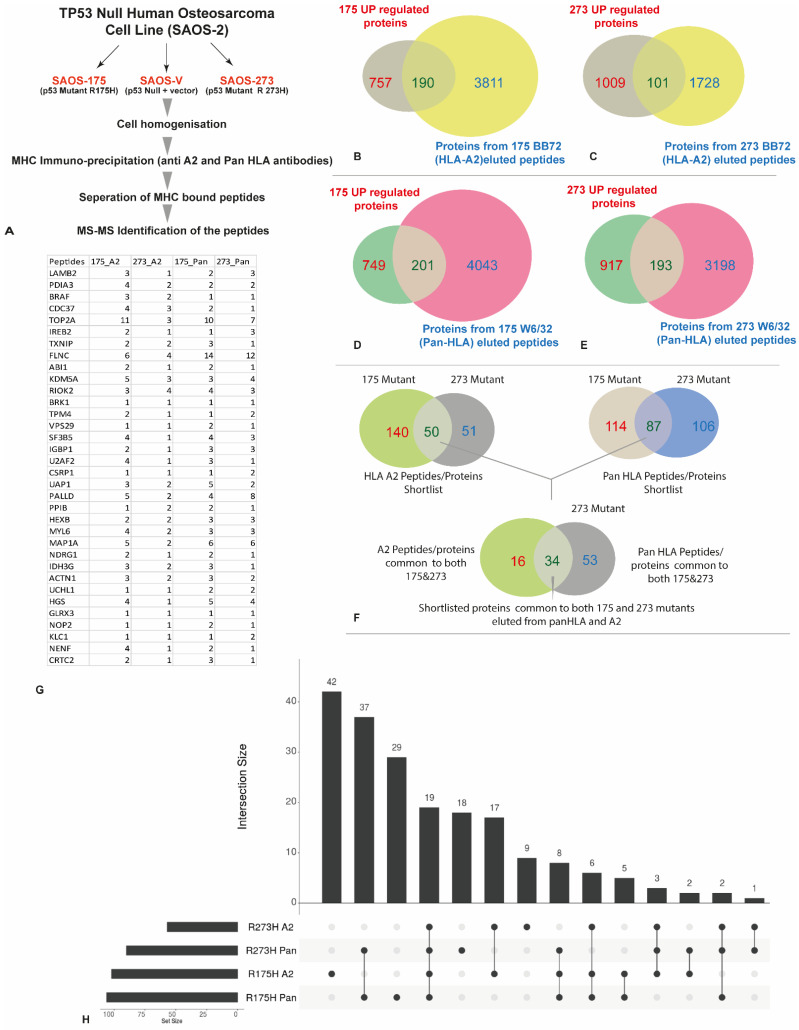
Immunopeptidome profiling of two *TP53* mutants revealed distinct HLA-A2 and pan-HLA peptide repertoire. (**A**) Workflow followed for generating MHC peptidome data from the three SaOS variants; (**B**) Venn diagram showing number of common proteins upregulated and HLA-A2 epitopes identified in 175 mutants; (**C**) Venn diagram showing number of common proteins upregulated and HLA-A2 epitopes identified in 273 mutant; (**D**) Venn diagram showing number of common proteins upregulated and Pan-HLA epitopes identified in 175 mutants; (**E**) Venn diagram showing number of common proteins upregulated and Pan-HLA epitopes identified in 273 mutant; (**F**) protein candidates common to both mutants and both eluted fractions (eluted by BB72 and W6/32); (**G**) List of 34 unique protein candidates and the number of epitopes identified in each mutants in both BB72 and W6/32 fractions; (**H**) upset plot showing unique and common peptide epitopes identified in each mutants in HLA-A2 and Pan HLA fractions.

**Figure 4 cancers-14-03975-f004:**
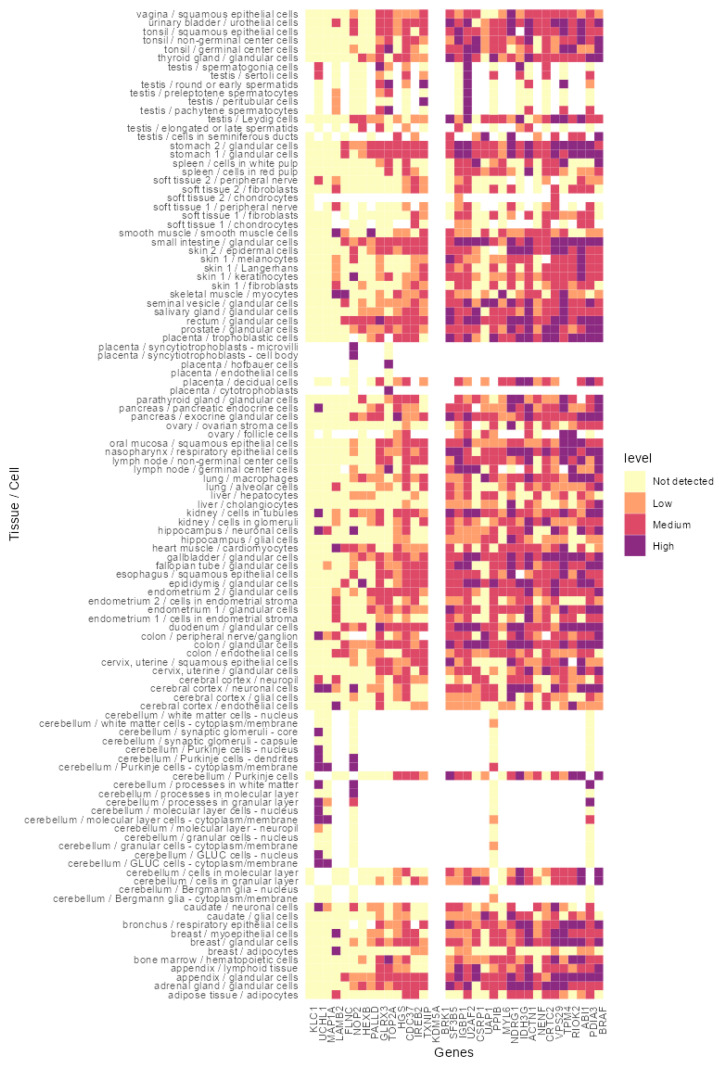
In silico analysis of the antigens revealed the levels of expression in normal tissues. Target antigens derived from p53-R175H and p53-R273H mutants and common to both HLA fractions (Pan-HLA and HLA-A2) were checked for their normal tissue expression. Heatmap showing the expression levels of each protein in normal cells. The purple, red and orange colors indicate very high to low levels of expression, pale yellow represents no expression detected, and white represents no data available in the corresponding histology in Human Protein Atlas.

**Figure 5 cancers-14-03975-f005:**
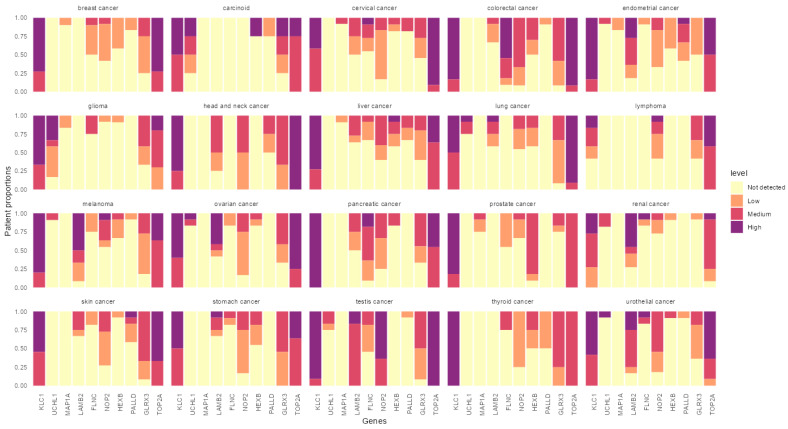
Expression of candidate antigens in twenty different tumor types. Tumor expression of shortlisted candidate genes/proteins identified both SaOS-R175H and SaOS-R273H cells in various cancer types. y-axes represent the proportion of patients expressing each candidate. The color of each bar indicates the strength of expression, with red being highly expressed and yellow as low expression.

**Table 1 cancers-14-03975-t001:** Number of HLA class I peptides (8–15 mers) identified across transfected SaOS cell lines.

Condition	HLA-A2 (BB7.2) Peptides	Pan HLA (W632) Peptides	Total
No vector control	2688	8982	11,670
R175H	9127	9814	18,941
R273H	2970	7145	10,115

## Data Availability

All data generated during this study are openly accessible at doi:10.5281/zenodo.5651927. All the immunopeptidomics data has been deposited to ProteomeXchange with project accession number PXD03044 with Reviewer account details: Username: reviewer_pxd034044@ebi.ac.uk and Password: NSjphEUx.

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
