# Peer review of "Multi-Omic Analysis of Two Common P53 Mutations: Proteins Regulated by Mutated P53 as Potential Targets for Immunotherapy"

_cancers, 2022, doi:10.3390/cancers14163975_

Round 1
Reviewer 1 Report
The authors performed multi-omics analyses on two P53 mutants in P53 null cells and identified several targets of immunotherapy. I think the manuscript is well written in general. I just had a suggestion:
It will be nice to see western blots on mutant P53 expressions in transfected cells or some functional assay to confirm that the P53 and its mutants were expressed and at what level.
I have no concern about the quality of the work and I would recommend acceptance of the manuscript subject to the correction of minor errors.
Author Response
Reviewer 1:
The authors performed multi-omics analyses on two P53 mutants in P53 null cells and identified several targets of immunotherapy. I think the manuscript is well written in general. I just had a suggestion:
It will be nice to see western blots on mutant P53 expressions in transfected cells or some functional assay to confirm that the P53 and its mutants were expressed and at what level.
Thank you. Confirmation of p53 expression levels was assessed by SWATH-MS in the mutants versus null vector. This is shown in Figure 2C.
I have no concern about the quality of the work and I would recommend acceptance of the manuscript subject to the correction of minor errors.
Thank you again.

Reviewer 2 Report
In this study, two mutant p53 (R175H and R273H) osteosarcoma cell lines are created to study TP53 mutations induced changes in gene expression, proteomic and immunopeptidome profiling. Mutant p53 upregulated and downregulated mRNA and proteins are comprehensively discussed. Furthermore, Immunopeptidome analysis revealed unique HLA-A2 and pan-HLA peptides in both mutant lines. At last, in silico analysis narrowed down the candidate antigens that are overexpressed in a variety of cancers compared to normal tissues. In conclusion, they have successfully provided a shortlist of potential immunotherapy targets for treating cancers caused by TP53 mutations. This is an interesting study exploring the secondary targets after TP53 mutations, however, there are some concerns that should be addressed before publication.
1. The data of Fig.1 on mRNA levels should be compared to data of Fig.2 on protein levels, so as to reveal the consistency between these two methods (mRNA array and SWATH proteomics)
2. In the analysis of Fig.4 and 5, it’s better to include the p53 status (wild-type, missense or null), since the candidate targets are screened by comparing missense mutations (R175H and R273H) versus null (SaOS-2-V). It’s shown that the candidate genes are more expressed in cancers compared to normal tissues, but it’s unclear whether they will have a positive correlation with TP53 mutations.
3. Since all the experiments are done in osteosarcoma cell line SaOS-2, there should be some discussion with respect to osteosarcoma.
4. In line 257, it says Fig.1D is colony formation assay, but in the Figure legend (line 312), it says Fig.1D is cell adhesion assay. Which one is right? If it’s not colony formation assay, representative images should also be provided.
5. In lines 54 and 55: it would be more accurate to use “R175, G245, R248, R249, R273 and R282”, as G245 is usually mutated to S or D, while R248 is usually mutated to Q or W. These hotspot mutations happens in various tumor types, for example, R249S is the sole hotspot mutation in hepatocellular carcinoma, so “in breast cancers, soft tissue, and bone sarcoma” should be replaced by a more accurate statement, like “in various cancers”.
6. In lines 64-66, it says R175H belongs to both “conformational mutant and DNA contact mutant”, which is not possibly true. According to Dr. Prives’s review paper (Reference 3), R175, G245 and R249 belong to conformational mutants, while R248, R273 and R282 belong to DNA contact mutants.
7. In line 251, “encoding conformation the p53 mutant R175H” should be “encoding the conformational p53 mutant R175H”.
8. In lines 266 and 267, “34 genes were commonly up regulated... compared to the SaOS-V vector control” should be moved to line 263 before “(Figure 1H and I; Supplementary Table 1).”
9. In line 451 it says “Levine’s group showed...”, but the “reference 3” cited is actually from Prives’ group.
10. In the plot legend of Fig.2D, after the orange dot, “p-value and log2” should be “p-value and log2FC”
11. In Fig.2F, some words on the left are missing.
12. In Fig.4, the plot legend is incorrect (white color not included) according to the figure legend.
13. When p53 is referred to as a gene, it should be TP53. Like in line 100, it should be “TP53 mutation status was confirmed...”. Also, all “TP53” should be italic in this manuscript.

Author Response
Reviewer 2:
In this study, two mutant p53 (R175H and R273H) osteosarcoma cell lines are created to study TP53 mutations induced changes in gene expression, proteomic and immunopeptidome profiling. Mutant p53 upregulated and downregulated mRNA and proteins are comprehensively discussed. Furthermore, Immunopeptidome analysis revealed unique HLA-A2 and pan-HLA peptides in both mutant lines. At last, in silico analysis narrowed down the candidate antigens that are overexpressed in a variety of cancers compared to normal tissues. In conclusion, they have successfully provided a shortlist of potential immunotherapy targets for treating cancers caused by TP53 mutations. This is an interesting study exploring the secondary targets after TP53 mutations, however, there are some concerns that should be addressed before publication.
- The data of Fig.1 on mRNA levels should be compared to data of Fig.2 on protein levels, so as to reveal the consistency between these two methods (mRNA array and SWATH proteomics)
P53 mRNA expression (Fig 1B) was performed using conventional PCR to confirm the successful transfection. The results show a higher mRNA expression of p53 by R273H than R175H (Fig 1B) which correlates with the protein expression as assessed by mass spectrometry in Fig 2C. Lines 287-8.
- In the analysis of Fig.4 and 5, it’s better to include the p53 status (wild-type, missense or null), since the candidate targets are screened by comparing missense mutations (R175H and R273H) versus null (SaOS-2-V). It’s shown that the candidate genes are more expressed in cancers compared to normal tissues, but it’s unclear whether they will have a positive correlation with TP53mutations.
Indeed, you make a good point that ideally we would know the p53 status for the tissues/cancer. However these data are not available via the Human Tissue Atlas database which is where the expression levels were generated for the proteins of interest.
- Since all the experiments are done in osteosarcoma cell line SaOS-2, there should be some discussion with respect to osteosarcoma.
This cell line (SaOS-2) is naturally p53null/null and is commonly used as a classic model to study p53 following transfection. We have used it as a model to assess the effect/influence of mutant p53 on the protein and gene expression profile, not osteosarcoma specific.
- In line 257, it says Fig.1D is colony formation assay, but in the Figure legend (line 312), it says Fig.1D is cell adhesion assay. Which one is right? If it’s not colony formation assay, representative images should also be provided.
Corrected.
- In lines 54 and 55: it would be more accurate to use “R175, G245, R248, R249, R273 and R282”, as G245 is usually mutated to S or D, while R248 is usually mutated to Q or W. These hotspot mutations happens in various tumor types, for example, R249S is the sole hotspot mutation in hepatocellular carcinoma, so “in breast cancers, soft tissue, and bone sarcoma” should be replaced by a more accurate statement, like “in various cancers”.
Corrected.
- In lines 64-66, it says R175H belongs to both “conformational mutant and DNA contact mutant”, which is not possibly true. According to Dr. Prives’s review paper (Reference 3), R175, G245 and R249 belong to conformational mutants, while R248, R273 and R282 belong to DNA contact mutants.
We could not find this statement on lines 64-66, as the text here did state they are different (conformational vs DNA binding), however on line 87 there was a typographical error that has now been corrected so as not to imply that both mutants are conformational.
- In line 251, “encoding conformation the p53 mutant R175H” should be “encoding the conformational p53 mutant R175H”.
Done
- In lines 266 and 267, “34 genes were commonly up regulated... compared to the SaOS-V vector control” should be moved to line 263 before “(Figure 1H and I; Supplementary Table 1).”
Done
- In line 451 it says “Levine’s group showed...”, but the “reference 3” cited is actually from Prives’ group.
Yes, this is correct, and this has been changed now
- In the plot legend of Fig.2D, after the orange dot, “p-value and log2” should be “p-value and log2FC”
p-value and log2FC added in figure 2D
- In Fig.2F, some words on the left are missing.
All the words are visible now
- In Fig.4, the plot legend is incorrect (white color not included) according to the figure legend.
Thank you, we have now corrected the legend accordingly
- When p53 is referred to as a gene, it should be TP53. Like in line 100, it should be “TP53mutation status was confirmed...”. Also, all “TP53”should be italic in this manuscript.
This has been corrected throughout the manuscript now.

Reviewer 3 Report
Attached.

Author Response
Reviewer 3:
Summary: Authors used two p53 mutants of SaOS-2 cell line to examine phenotypic and functional changes utilizing different techniques in combination with genomics and proteomics. Research suggests few candidate genes that could be potentially examined as immunotherapy targets after additional in-vivo validation.
Major comment: p53 is mutated in wide variety of cancer. Why use single cell line to study mutated p53? Why SaOS-2 cells in specific? Suggestion: It is recommended to provide the justification of using only single cell line in this study. Also, the rationale behind using SaOS-2 cell line must be explained.
This has been explained on lines 91-92.
General comment: In introduction section, briefly mentioning other broader approaches to target p53 as potential therapy is recommended.
Specific comments:
Specific comment 1: In line 261, it is mentioned corrected p Value of 0.01? Does it mean adjusted p value for multiple testing?
Yes this is correct
Suggestions: It is recommended to use the term adjusted p-values for multiple comparison as it sounds more scientific that these values were adjusted for multiple comparison.
Thank you. Suggestion has been implemented.
Specific comment 2: Figure 3H: There is a better way to represent this information. The colors and circles are not easier for readers to interpret. Suggestions: Alternative to Venn diagram, it is recommended to use “static Upset plots” to represent the data for making it easier and more comprehensive to interpret. For e.g.: There is a R package called UpSetR which will generate better visualization.
Thank for this and we have incorporated this to replace the four-way Venn diagram in figure 3
Specific comment 2b: Similar changes suggested in comment 2 can be made for Figure 1 (H and L).
We have decided to leave the two-way Venn diagram (Figure 1 (H and L)) as it is easy to follow
Specific comment 3: Line 67 needs proper citation for “abolish the contact of the protein with its DNA target”
This is incorporated in reference 3 (Line 64)
Editor:
Please provide details on transfection protocol. Sanger sequencing electropherogram should be introduced in fig 1C to confirm the single point mutation, respectively, for R175H and R273H Check figures and tables (typing errors or cutting figures are present, i.e. fig. 2F).
Transfection protocol is +-available in reference 15
Sanger sequencing traces added to figure 1 showing point mutations
